# A Novel, Simple and Highly Efficient Route to Obtain PrBaMn_2_O_5+δ_ Double Perovskite: Mechanochemical Synthesis

**DOI:** 10.3390/nano11020380

**Published:** 2021-02-02

**Authors:** Francisco J. Garcia-Garcia, María J. Sayagués, Francisco J. Gotor

**Affiliations:** 1Departamento de Ingeniería y Ciencia de los Materiales y del Transporte, Universidad de Sevilla, E-41092 Seville, Spain; 2Instituto de Ciencia de Materiales de Sevilla (CSIC-US), E-41092 Seville, Spain; mjsayagues@icmse.csic.es (M.J.S.); francisco.gotor@icmse.csic.es (F.J.G.)

**Keywords:** SOFCs, mechanochemistry, layered double perovskites, sinterability, electrical conductivity

## Abstract

In this work, a mechanochemical route was proposed for the synthesis of the PrBaMn_2_O_5+δ_ (PMBO) double layered perovskite phase. The mechanochemical reaction between Pr_6_O_11_, BaO_2_, and MnO powders with cationic stoichiometric ratios of 1/1/2 for Pr/Ba/Mn was performed using high-energy milling conditions in air. After 150 min of milling, a new phase with perovskite structure and cubic symmetry consistent with the A-site disordered Pr_0.5_Ba_0.5_MnO_3_ phase was formed. When this new phase was subsequently annealed at a high temperature in an inert Ar atmosphere, the layered PrBaMn_2_O_5+δ_ phase was obtained without needing to use a reducing atmosphere. At 1100 °C, the fully reduced layered PrBaMn_2_O_5_ phase was achieved. A weight gain was observed in the 200–300 °C temperature range when this fully reduced phase was annealed in air, which was consistent with the transformation into the fully oxidized PrBaMn_2_O_6_ phase. The microstructural characterization by SEM, TEM, and HRTEM ascertained the formation of the intended PrBaMn_2_O_5+δ_ phase. Electrical characterization shows very high electrical conductivity of layered PBMO in a reducing atmosphere and suitable in an oxidizing atmosphere, becoming, therefore, excellent candidates as solid oxide fuel cell (SOFC electrodes).

## 1. Introduction

Solid oxide fuel cells (SOFCs) are one of the most promising and environmentally friendly technologies for generating electricity from a variety of fuels. SOFCs directly transform chemical energy into electricity without the need for combustion (electrochemical conversion), resulting in much higher efficiencies, since losses derived from thermodynamic considerations are minimized [1,2]. If hydrogen is employed as fuel, only water is generated as a byproduct of the process. In addition, as SOFCs operate at high temperatures, 800–1000 °C, they are capable of producing hydrogen inside the cell by internal reforming from natural gas and other fuels which are easier to handle and transport than hydrogen [3,4]. However, operating directly with hydrocarbons produces carbon deposition and sulfur poisoning in conventional Ni-based cermet anodes [5,6,7], and therefore deactivation, which requires desulfurizing and/or external reforming units [8], with the consequent increase in the complexity of the design and the stack costs.

This is one of the reasons why the search for alternative anode materials resistant to carbon and sulfur is receiving special and continuous attention [9]. Oxides with perovskite structure (with general formula ABO_3_) have been proposed as candidates for anodes, as they are flexible in composition and allow a high degree of cationic substitution in A- and B-sites, which permits the tailoring of the final properties, such as the electronic and ionic conductivities, by properly adjusting the stoichiometry. The presence in the B-site of transition metals with multiple oxidation states is of primary importance because it largely determines the electrocatalytic properties and stability upon redox cycling [10,11]. The different perovskite materials proposed to date as anodes can be found in several reviews [12,13], in which their main advantages and disadvantages are highlighted.

Double perovskites with the general formula A_2_BB′O_6_, such as Sr_2_CoMoO_6-δ_, Sr_2_MgMoO_6-δ_, or Sr_2_TiMoO_6-δ_ [14,15,16], have also been investigated as anode materials, but surface degradation, non-negligible reactivity with electrolytes, especially YSZ, and low power densities have been frequently observed. Recently, double layered perovskite oxides with different A-site compositions (AA′B_2_O_6_), and particularly double layered manganites, including PrBaMn_2_O_5+δ_ (PBMO), have been proposed as promising alternatives to Ni/YSZ, not only as potential anode materials in SOFCs [17], but also as cathode materials in solid oxide electrolysis cells (SOECs) [18], due to their interesting properties, such as high electrical conductivity (mixed ionic and electronic), fast surface oxygen exchange, and easy oxygen anion transport [19].

In PBMO structure, the Ba and Pr cations are not distributed randomly in the perovskite A-site, but ordered in alternating Pr and Ba layers along (001). All the oxygen vacancies in PBMO are located in the Pr A-site layers [20], which plays a key role in creating fast oxygen ion diffusion paths. It has been shown that under SOFC anode conditions, the oxygen site in the Pr-O plane is vacant or near-vacant (δ~0) [20]. Furthermore, PBMO has proven to be very stable over a wide range of temperatures and oxygen partial pressures [21]. In this sense, PBMO has shown almost complete reversibility between the fully-reduced (δ~0) and fully-oxidized (δ~1) phases at relatively low temperatures (200–500 °C) when the oxygen partial pressure is alternated between reducing and oxidizing conditions [21,22]. Moreover, it has also been proven to withstand redox cycles at higher temperatures (up to 950 °C) with only a small decrease in the maximum oxygen storage capacity [23]. These features are due to the high order degree in A-sites, which provides thermal and chemical stability under operating conditions, and the ability of the transition metal cation to accommodate different valence states (Mn^4+^/Mn^3+^/Mn^2+^), which favor the intake and release of oxygen within the vacancy rich Pr A-site layer during the redox processes.

PBMO has shown an electrical conductivity of 8.16 S/cm at 800 °C in 5% H_2_ [17]. In an electrolyte-supported single cell with a configuration of PBMO/La_0.4_Ce_0.6_O_2-δ_ (LDC)/La_0.9_Sr_0.1_Ga_0.8_Mg_0.2_O_3-δ_ (LSGM)/NdBa_0.5_Sr_0.5_Co_1.5_Fe_0.5_O_5+δ_-Ce_0.9_Gd_0.1_O_2-δ_ (NBSCF50-GDC), a maximum power density of 0.66 W/cm^2^ was obtained at 800 °C in humidified H_2_ (3% H_2_O) [24]. An LDC buffer layer is frequently added to avoid PBMO chemical reactivity with the electrolyte material at high temperature [17,24]. Although Mn-containing perovskites generally have reasonable activity for hydrocarbon oxidation, for practical applications, the electrocatalytic activity of the PBMO anode should be further enhanced by adding active metal catalysts. For example, power densities of 1.7 W/cm^2^ were obtained at 850 °C in humidified H_2_ when PBMO with a 15 wt.% Co-Fe catalyst was used, with no observable degradation in H_2_S/H_2_ [17]. This same anode also showed good coking resistance in humidified propane with a maximum power density of 1.3 W/cm^2^ at 850 °C [17]. The exsolution of catalyst nanoparticles on the surface of the PBMO anode material under reducing conditions (during SOFC anode operation) when the B-site is doped with catalytically active transition metals (Co, Fe, Ni) has also shown promising results regarding the electrode performance and long-term stability [24,25].

PBMO powders are generally prepared by a two-step synthesis procedure. First, a disordered Pr_0.5_Ba_0.5_MnO_3_ precursor is obtained from a sol-gel route using EDTA or citric acid as the chelating agent, which includes a final pyrolysis step at high temperature in air [17,20,24,25,26,27], or by a conventional solid-state reaction method [18]. Note that the obtained Pr_0.5_Ba_0.5_MnO_3_ precursor is generally composed of a mixture of two phases with cubic and hexagonal structures. The major cubic phase corresponds to the A-site disordered perovskite, while the minor hexagonal phase is associated with the BaMnO_3_ secondary phase, whose formation is favored by a high extent of oxidation from Mn^3+^ to Mn^4+^ under the experimental conditions of synthesis. In a second step, the Pr_0.5_Ba_0.5_MnO_3_ precursor is annealed at ~800–900 °C in a reducing atmosphere, during which the hexagonal structure is suppressed, and the A-site ordered layered PBMO is formed with a tetragonal symmetry.

Mechanochemistry has shown to be an efficient, reproducible, and relatively simple way to obtain mixed oxides with simple perovskite structure that can be used as components in SOFCs [28,29,30,31,32,33]. This methodology is based on the induction of solid-state reactions at room temperature by applying mechanical energy (instead of heating) to a reactant powder mixture using primarily high-energy ball milling equipment. In recent years, there has been a strong revitalization of mechanochemistry, as it can be applied to a wide variety of materials, from metals and alloys to organic compounds [34,35]. Note that by properly adjusting the milling conditions, the contamination from the milling media, which has been one of the main drawbacks for which mechanochemistry has been criticized for years, can be reduced to low and permissible levels for most technological applications [36]. In addition, the use of solvents can also be avoided and, therefore, the advantage of not producing waste during preparation constitutes an additional benefit from an environmental point of view.

Several studies can be found in the literature dealing with the mechanochemical synthesis of selected layered perovskite oxides, such as the Ruddlesden-Popper (A_n+1_B_n_O_3n+1_), Aurivillius (Bi_2_O_2_) (A_n−1_B_n_O_3n+1_) or double perovskites (A_2_BB′O_6_) phases, but in general, an annealing step after the milling treatment of the reactant mixture was required to obtain the intended final layered phase [37,38,39,40,41,42,43]. Regarding the double layered perovskite oxides, milling procedures have only been used to prepare composite cathodes, such as, for example, PrBa_0.92_Co_2_O_6-δ_-Ce_0.8_Sm_0.2_O_1.9_, from the different phases previously synthesized by other methods [44], or to improve the oxygen storage properties of BaPrMn_2_O_5+δ_ and BaSmMn_2_O_5+δ_ previously obtained at high temperatures [45].

In this work, the direct mechanochemical synthesis of the double layered PBMO material by high-energy milling of the mixture of the corresponding binary oxides using a planetary mill is proposed. The evolution of the PBMO conversion with the milling time was studied and the chemical, structural, microstructural, and electrical properties of the final product were fully characterized.

## 2. Materials and Methods

Praseodymium (III,IV) oxide (Pr_6_O_11_, CAS Number 12037-29-5, Alfa Aesar, 99% in purity), barium peroxide (BaO_2_, CAS Number 1304-29-6, Aldrich, 95% in purity) and manganese (II) oxide (MnO, CAS Number 1344-43-0, Alfa Aesar, 99% in purity) powders were used as the starting reactants for the mechanochemical synthesis of PrBaMn_2_O_5+δ_. For the experiments, the stoichiometric amounts of the powder reactants to produce 5 g of PrBaMn_2_O_5+δ_ and six tungsten carbide (WC) balls (*d* = 15 mm and *m* = 26.4 g) were placed in a 60 mL tempered steel vial and milled in air in a planetary mill (Micro Mill Pulverisette 7, Fritsch) at a spinning rate of 600 rpm. The milling process was stopped at 15 min intervals for X-ray diffraction (XRD) inspection. A total milling time of 150 min was used.

The XRD patterns were collected on an X’Pert Pro MPD diffractometer (PANalytical) equipped with a θ/θ goniometer, a graphite-diffracted beam monochromator and a solid-state detector (X’Cellerator). The diffraction patterns were acquired using Cu Kα radiation from 10° to 140° (2θ) in step-scan mode with a step size of 0.05° and a counting time of 300 s/step. The diffraction line positions were corrected with silicon powder (Standard Reference Material 640c, NIST). Phase quantification, lattice parameters, crystalline domain sizes (D) and microstrains (e) were calculated from the Rietveld refinement tool offered by the X’Pert HighScore Plus software (Version 3.0.5, PANalytical) using pseudo-Voigt functions to describe the line profiles.

For the microstructural characterization, the powder samples were dispersed in acetone and droplets of the suspension were deposited onto a holey carbon copper grid. The scanning electron microscopy (SEM) images were obtained on a Hitachi S-4800 SEM-FEG microscope in secondary electron mode at an acceleration voltage of 5 kV. Energy-dispersive X-ray analysis (EDX) was performed with a detector coupled to the SEM microscope using an acceleration voltage of 20 kV. The transmission electron microscopy (TEM) and high-resolution (HRTEM) images, selected area electron diffraction (SAED) patterns, and EDX spectra were taken on a 200 kV JEOL-2100-PLUS microscope equipped with a LaB_6_ filament (point resolution = 0.25 nm). HR micrograph analysis, fast Fourier transform (FFT), and phase interpretation were performed with the Gatan Digital Micrograph software (Gatan Inc.) and the Java version of the electron microscope software (JEMS).

The thermogravimetric measurements were carried out in a simultaneous thermal analysis equipment (STA 449 F5 Jupiter, Netzsch) on 60–80 mg of powders using an alumina sample pan at a heating rate of 10 °C/min from room temperature up to 1100 °C in two different flowing atmospheres, air (50 mL/min) and Ar (50 mL/min).

Impedance spectra of reduced and oxidized layered PrBaMn_2_O_5+δ_ were measured using an impedance analyzer model Solartron 1260A at open circuit voltage (OCV). All measurements were made at from 800 to 100 °C, with a range of 100 °C. The specimens were measured in synthetic airflow (150 mL/min) and Ar/10%H2 (150 mL/min) with a frequency range of 1 MHz–0.1 Hz. All the spectra were analyzed with the ZView software. Platinum wires were used as voltage and current collectors in a single chamber configuration. For the impedance measurements, 0.6 g of the powder was compacted in a disk shape (*∅* = 12 mm and thick = 1 mm) using a uniaxial press at 100 MPa for 5 min and further sintered at 1500 °C in argon for 12 h, at heating and cooling rates of 5 °C/min, in a tubular furnace. Finally, the specimens were ground and polished to achieve a final thickness of 1 mm.

## 3. Results and Discussion

A Pr_6_O_11_/BaO_2_/MnO powder mixture with a stoichiometric ratio according to Equation (1) was milled using the experimental conditions described in the Materials and Methods section.
(1)$$\frac{1}{6}{\mathrm{Pr}}_{6}O_{11}\;+{\;\mathrm{BaO}}_{2}\;+2\mathrm{MnO}\;\to {\;\mathrm{PrBaMn}}_{2}O_{5+\delta }$$

The milling process was stopped every 15 min and the powder was inspected by XRD. Figure 1 shows the XRD patterns of the mixture as a function of the milling time. The XRD pattern at 0 min corresponds to the starting reactant mixture. Figure 1 clearly shows how the intensity of the XRD peaks of reactants (especially BaO_2_) was considerably reduced after only 15 min of milling. Moreover, the XRD peaks also suffered from a large broadening. Both effects were the result of the high-intensity milling regime employed. At this short time of 15 min, new XRD peaks began to develop suggesting the formation of a new phase with a perovskite structure. With increasing milling time, the mechanochemical reaction progressed and the amount of this new phase continuously increased. After 150 min, the conversion was finished and the mechanochemical solid state reaction seemed to be nearly complete (from now this sample will be named PBMO-m). Only the existence of a minor amount of BaCO_3_, which was already present in the starting BaO_2_ as a minor phase, was observed. The percentage of the newly formed perovskite phase as a function of the milling time calculated from the XRD patterns in Figure 1 is plotted in Figure 2.

The XRD peaks of the new perovskite phase show extremely large broadening due to the small size of the coherent diffraction domains (*D* = 14.3 nm) and the presence of a high number of defects (microstrain, *e* = 0.434%) as a result of the high-intensity milling conditions employed. The XRD pattern was consistent with the A-site disordered Pr_0.5_Ba_0.5_MnO_3_ phase with cubic symmetry and space group *Pm-3m* (*a* = 3.8962(3) Å); the corresponding (*h k l*) are marked in Figure 1. However, a similar cubic phase has also been observed in perovskite materials with actual lower symmetry structure when obtained by mechanochemistry [46]. Because of the large broadening of the XRD peaks, the real structure of the perovskite phases obtained by mechanochemistry is hardly resolved, and they always appear with a pseudo-cubic symmetry. Note that XRD patterns similar to that observed in Figure 1 for the PBMO-m sample were assigned in the literature to the layered PBMO phase [17,24]. In this sense, lattice parameters *a* = 5.522(1) Å and *c* = 7.757(3) Å were obtained from this XRD pattern (PBMO-m sample), assuming the tetragonal structure with space group *P4/nmm* characteristic of the reduced-form of the PrBaMn_2_O_5+δ_ phase. Furthermore, the presence of the hexagonal phase associated with the BaMnO_3_ secondary phase, as generally found in thermal-related synthesis processes, was not detected in this mechanochemical route.

To better crystallize the structure and resolve the real crystal symmetry of the perovskite phase obtained by mechanochemistry, the PBMO-m sample was annealed for 3 h at increasing temperatures in an inert Ar atmosphere. Figure 3 shows the room temperature XRD patterns of samples after annealing (500–1100 °C range). Note that after the thermal treatment at 500 °C, BaCO_3_ impurity phase disappeared. It can be also observed that the broadening of the XRD peaks slightly decreased at increasing annealing temperature as a result of a better crystallinity. Moreover, the XRD peaks shifted to lower angles, indicating larger lattice parameters in annealed samples. The pseudo-cubic symmetry seemed to be maintained up to 800 °C. However, the larger broadening observed in the XRD peak located at ~46° 2θ in the annealed sample at 800 °C compared to the same peak at 700 °C suggests that at 800 °C the cubic structure has started to evolve toward the tetragonal symmetry. At 900 °C, the tetragonal structure characteristic of the PrBaMn_2_O_5+δ_ phase (*P4/nmm*) was already well defined. The lattice parameters determined for this sample, labelled as PBMO-m + 900 °C/Ar, were *a* = 5.6179(4) Å and *c* = 7.7695(7) Å. Annealing at higher temperature (1100 °C) did not produce any new significant modification (just a smaller peak broadening) and the sample, labelled as PBMO-m + 1100 °C/Ar, maintained similar lattice parameters (*a* = 5.6219(5) Å and *c* = 7.7584(1)), which were in close agreement with the literature values determined for the fully reduced PrBaMn_2_O_5_ phase [21,26].

As mentioned above and according to the literature, the fully reduced PrBaMn_2_O_5_ phase can be obtained from the disordered Pr_0.5_Ba_0.5_MnO_3_ or the fully oxidized PrBaMn_2_O_6_ phases when they are annealed at 800–900 °C in a reducing atmosphere [17,18]. Moreover, it has been shown that while heating YBaMn_2_O_6_ and GdBaMn_2_O_6_ up to 980 °C under an inert atmosphere leads to the formation of the half-reduced phase with an oxygen stoichiometry close to 5.5, the same treatment for PrBaMn_2_O_6_ only produces a slight decrease in the oxygen content in the oxidized phase [21]. Therefore, it seems surprising that if the disordered Pr_0.5_Ba_0.5_MnO_3_ was obtained by mechanochemistry, the reduced layered phase was formed when the PBMO-m sample was annealed (above 800 degrees) in an inert atmosphere.

To obtain new insights into the nature of the perovskite phase obtained by mechanochemistry and to estimate the oxygen content before and after the annealing treatments in Ar, thermogravimetric measurements were performed in different atmospheres. The main results are depicted in Figure 4.

A weight loss of ~4.8% was observed when the PBMO-m sample was heated up to 850 °C in a flowing Ar atmosphere. This weight loss (Figure 4a) is even greater than the theoretical loss corresponding to 1 formula unit of oxygen associated with the transformation of Pr_0.5_Ba_0.5_MnO_3_ to PrBaMn_2_O_5_ (~3.3 wt.%) and it is the consequence of the presence in the product of BaCO_3_, which decomposes in the temperature range studied. However, when the PBMO-m + 900 °C/Ar and PBMO-m + 1100 °C/Ar samples were heated in air, a weight gain was observed in the 200–300 °C temperature range (Figure 4b,c, respectively) according to the oxygen uptake associated with the transformation of the reduced PrBaMn_2_O_5+δ_ phase into the oxidized PrBaMn_2_O_6-δ_ phase. If we assume that after the thermogravimetric measurements in air the layered perovskite is fully oxidized, the weight gains of 2.75 and 3.26 wt.% determined are in agreement with an approximated oxygen stoichiometry of PrBaMn_2_O_5.20_ and PrBaMn_2_O_5.05_ for the PBMO-m + 900 °C/Ar and PBMO-m + 1100 °C/Ar samples, respectively. Therefore, it is clear that the reduced layered structure (PrBaMn_2_O_5+δ_) was obtained when PBMO-m was annealed in Ar; the use of a reducing atmosphere is not necessary.

In Figure 5, the XRD patterns of the PBMO-m (Figure 5a) and PBMO-m + 1100 °C/Ar samples are compared with that of the fully oxidized layered phase obtained after the thermogravimetric measurements in flowing air (Figure 5c), labelled as PBMO-m **+** 1100 °C/Ar + TG/air. The XRD pattern of this last sample was consistent with a tetragonal *P4/mmm* structure with lattice parameters *a* = 3.9004(1) Å and *c* = 7.7582(2) Å in agreement with the literature values for PrBaMn_2_O_6_ [21,26].

A microstructural characterization was carried out for the PBMO-m, PBMO-m + 1100 °C/Ar and PBMO-m + 1100 °C/Ar + ATG/air samples. According to SEM micrographs obtained for the three samples (Figure 6), the morphological evolution can be easily observed after the mechanochemical and thermal treatments. The PBMO-m sample is formed by agglomerates of small rounded particles characteristic of powders obtained by dry mechanochemical reaction (Figure 6a). After annealing at 1100° C in Ar (PBMO-m + 1100 °C/Ar sample), the size of small particles and crystalline domains (as shown the XRD analysis) increased, displaying a wide range of sizes between 300 and 700 nm (Figure 6b) and a polygonal shape (inset). The PBMO-m + 1100 °C/Ar + ATG/air sample, which was obtained after the TG experiment of the previous one, presents quite similar morphology (Figure 6c); nevertheless, the crystal size, due to the TG heating, increased, exceeding 1 μm.

A TEM and related techniques study was carried out to deep into the microstructure of the three samples. The results obtained for sample PBMO-m are depicted in Figure 7. The crystalline domains form agglomerates (as was observed by SEM); however, the TEM image presented in Figure 7b shows the size of the independent crystallites between 10 and 20 nm, corroborating the figures found by XRD measurements. The nanometric character of the sample was also verified by the SAED pattern-rings (Figure 7c), where all of them could be indexed to the pseudo-cubic perovskite structure (*Pm-3m* (*a* = 3.8962(3) Å). The corresponding (*h k l*) planes are marked in the figure.

Figure 8 and Figure 9 present the results corresponding to the TEM and HRTEM characterization of the PBMO-m + 1100 °C/Ar and PBMO-m + 1100 °C/Ar + ATG/air samples. The difference in terms of symmetry between the tetragonal fully reduced layered phase (*P4/nmm*; 129) and the tetragonal fully oxidized layered phase (*P4/mmm*; 123) is very slight. They only diverge in a 45 degree rotation at the base of the structure in such a way that *a*_129_= √2*a*_123_ and the *c* parameter is nearly the same. For this reason, it is quite difficult to differentiate both structures by SAED. In the case of the PBMO-m + 1100 °C/Ar sample, several high-resolution micrographs were found, where the FFTs could be indexed to both structures. This fact can be seen when comparing FFTs with the electron diffraction ED calculated diagrams for both symmetries (Figure 8c,d). However, it was possible to find HRTEM images whose FFT could only be assigned to the tetragonal *P4/nmm* structure. A representative image is depicted in Figure 8e, oriented along the (3 1 1)_129_ zone axis and the (*h k l*) are marked. On the other hand, the microstructural analysis of the PBMO-m + 1100 °C/Ar + ATG/air sample (Figure 9), the oxidized layered phase, permitted to obtain HRTEM micrographs where the FFTs could only be indexed according to the *P4/mmm* tetragonal symmetry. The two examples shown in Figure 9c,d correspond to (4 2 1)_123_ and (2 1 1)_123_ zone axes, respectively. The (*h k l*) planes are marked in the figure. Therefore, it can be confirmed that the reduced and oxidized layered samples possess *P4/nmm* and *P4/mmm* tetragonal symmetry, respectively, in good agreement with the found XRD results.

The chemical composition (regarding the cations) of the three samples (PBMO-m, PBMO-m + 1100 °C/Ar and PBMO-m + 1100 °C/Ar + ATG/air) was analyzed by EDX spectroscopy and the corresponding spectra and average atomic percentage of the cations are presented in Figure 7d, Figure 8b and Figure 9b, respectively. Taking into account that the cation atomic percentage according to the stoichiometric formula of the PBMO is: Pr = Ba = 25% and Mn = 50%, the found experimental data are around these values with a maximum standard deviation of ±2.3%.

Figure 10 shows the electrical conductivity of PBMO-m after sintering at 1500 °C as a function of temperature in Ar/H_2_ and synthetic air from 100 to 800 °C. Remember that after the sintering process in Ar atmosphere, the reduced layered PrBaMn_2_O_5+δ_ phase was obtained. The electrical conductivity reaches a maximum of 21.1 S·cm^−1^ when reduced and 96.8 S·cm^−1^ in its oxidized state (measurement in air atmosphere). The very high electrical conductivity values reported for this layered PBMO double perovskite, particularly in air, are comparable to the values obtained by Sendogan et al. [17], indicating that the mechanochemical synthesis is an optimum method to obtain this phase, which can provide efficient electron transfer paths to meet the requirements of both the anode and cathode in SOFCs.

It is known [17,47] that the predominant defects in this type of double layered perovskites, such as PBMO, are mobile electronic holes (h^•^= Mn^•^_Mn_). According to the Kröger–Vink equation, the interaction between oxygen vacancies and mobile electronic holes can be expressed as follows
(2)2MnMnx+12O2+VO••  →← OOx+2MnMn•

This relationship implies that the concentration of oxygen vacancies will increase under a reducing atmosphere, yielding in enhanced oxide-ionic conductivity and oxygen ion transfer sites between the anode and the electrolyte. In an oxidizing atmosphere, the electrical charge is mainly balanced by the conversion of Mn^2+^ to Mn^3+^ and even to Mn^4+^ ions (although some Pr^4+^ may be present). In contrast, in a reducing atmosphere, the electroneutrality is maintained by the formation of oxygen vacancies followed by the reduction of Mn^4+^ to Mn^3+^ to Mn^2+^ ions.

The thermal dependence of conductivity for oxygen ionic conductors can be obtained from the well-known Arrhenius plot. The activation energy values of 0.06 and 0.33 eV were found in the oxidized and reduced specimens, respectively. The slopes obtained, and thus the activation energies calculated from the linear fits, are also consistent with data reported previously in layered PBMO double perovskites fabricated by other synthesis routes.

## 4. Conclusions

High-energy milling in air of a Pr_6_O_11_, BaO_2_ and MnO powder mixture with Pr/Ba/Mn stoichiometric ratios of 1/1/2 led to the formation of a pseudo-cubic perovskite phase consistent with Pr_0.5_Ba_0.5_MnO_3_. The presence of the hexagonal phase associated with the BaMnO_3_ secondary phase, as generally found in thermal-related synthesis processes, was not detected in this mechanochemical route, which yields in direct fabrication of the layered perovskite, without further annealing in an reducing atmosphere. If the perovskite phase obtained by mechanochemistry is annealed at an increasing temperature in an inert Ar atmosphere, the layered PrBaMn_2_O_5+δ_ phase is formed. This phase transforms into the oxidized layered PrBaMn_2_O_6-δ_ when treated in air at 200–300 °C. Electrical characterization shows very high electrical conductivity of layered PBMO, becoming therefore excellent candidates to be considered as SOFC electrodes.

## 5. Future Work

The fabrication of a full SOFC containing a PBMO anode or a symmetrical fuel cell with PBMO in both electrodes, as well as the characterization of such fuel cell performance is planned for future experiments. In addition, considering the benefits of this fabrication method and the good properties of the fabricated PBMO double perovskite, authors plan to incorporate other elements (B-site doping with transition metals) using this fabrication method in the double perovskite in order to further enhance the electrical properties.

## Figures and Tables

**Figure 1 nanomaterials-11-00380-f001:**
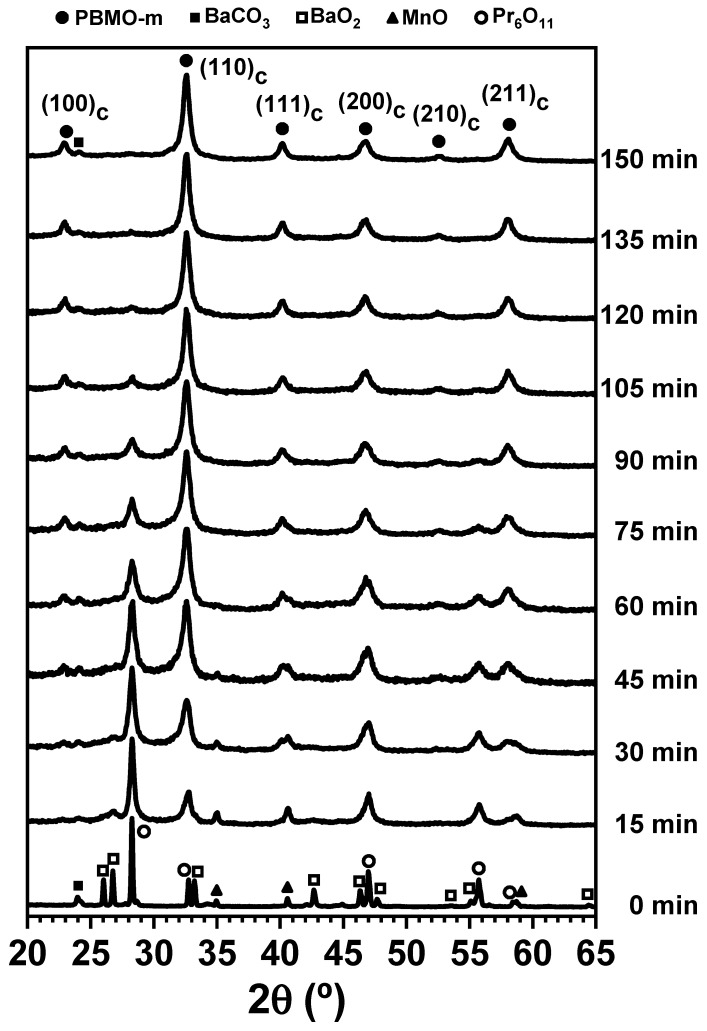
XRD patterns of the Pr_6_O_11_/BaO_2_/MnO powder mixture subjected at increasing milling time.

**Figure 2 nanomaterials-11-00380-f002:**
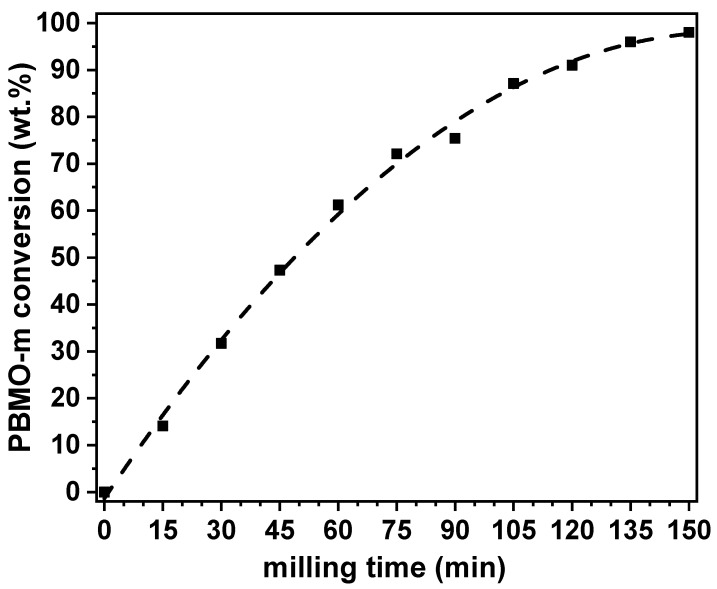
Quantification by Rietveld refinement of the XRD patterns in Figure 1 of the new perovskite phase formed when the Pr_6_O_11_/BaO_2_/MnO powder mixture is subjected at increasing milling time.

**Figure 3 nanomaterials-11-00380-f003:**
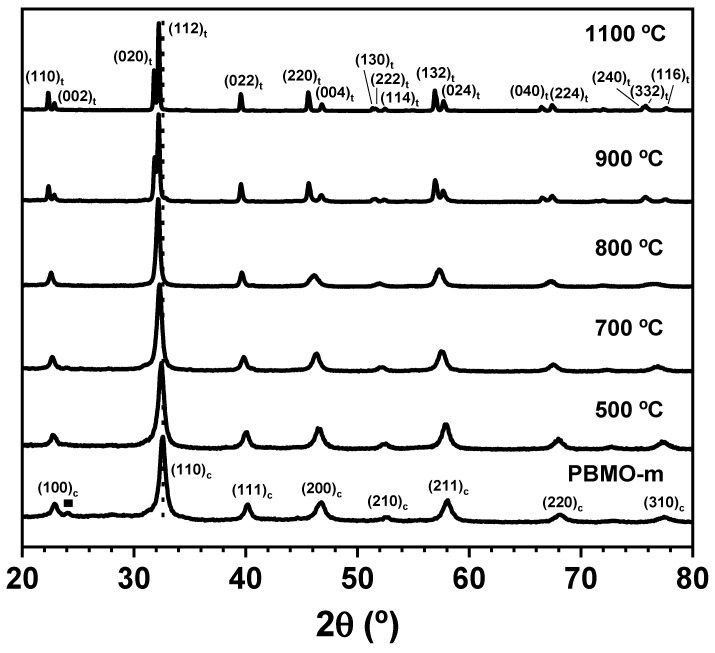
Room-temperature XRD patterns of PBMO-m sample after annealing for 3 h at increasing temperature in Ar. (▪ BaCO_3_).

**Figure 4 nanomaterials-11-00380-f004:**
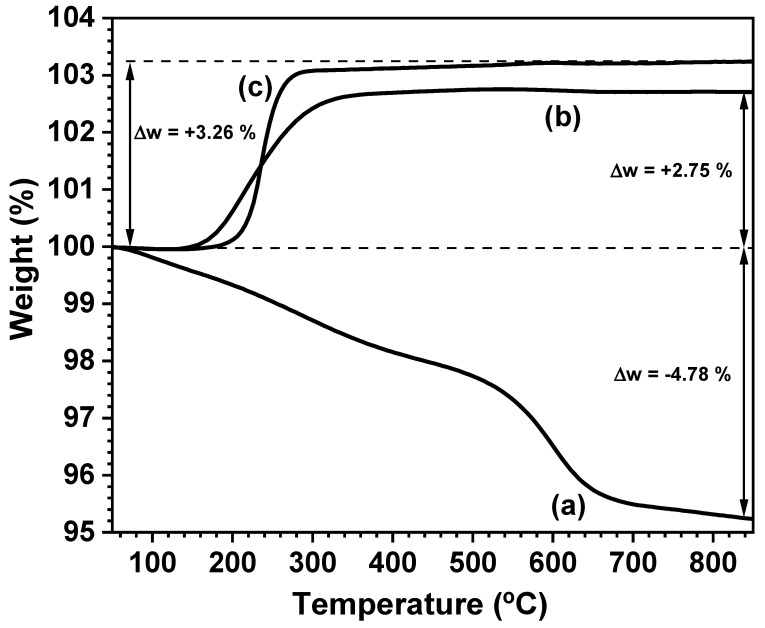
Thermogravimetric curves of (**a**) PBMO-m sample in flowing Ar, (**b**) PBMO-m + 900 °C/Ar sample in flowing air and (**c**) PBMO-m + 1100 °C/Ar sample in flowing air.

**Figure 5 nanomaterials-11-00380-f005:**
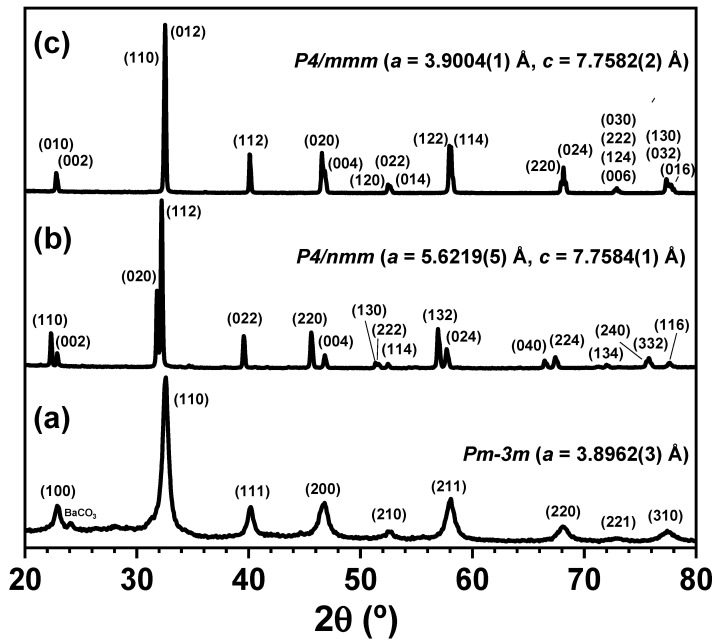
Room-temperature XRD patterns of (**a**) PBMO-m, (**b**) PBMO-m + 1100 °C/Ar and (**c**) PBMO-m + 1100 °C/Ar + TG/air samples.

**Figure 6 nanomaterials-11-00380-f006:**
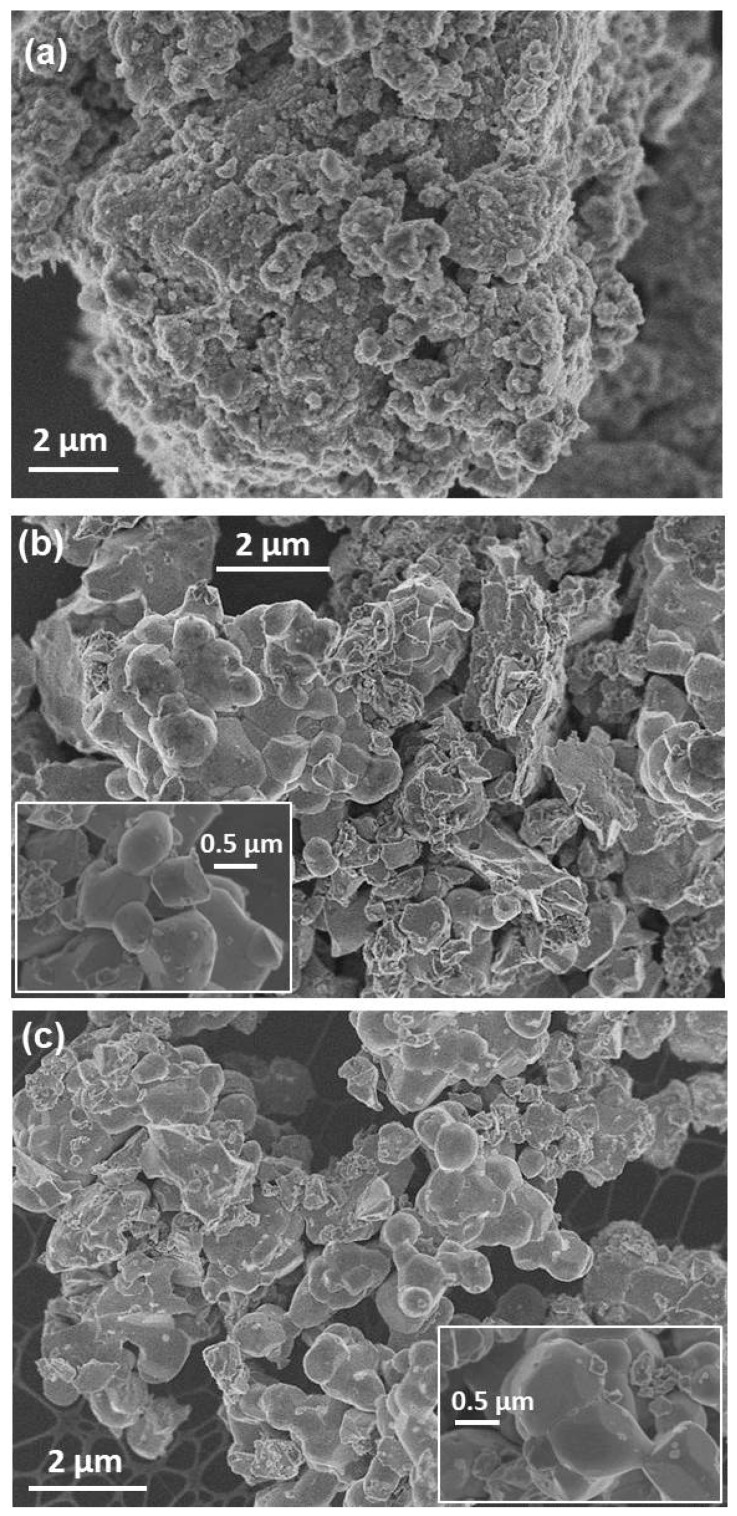
SEM micrographs of (**a**) PBMO-m, (**b**) PBMO-m + 1100 °C/Ar, and (**c**) PBMO-m + 1100 °C/Ar + TG/air samples. The insets correspond to higher magnifications.

**Figure 7 nanomaterials-11-00380-f007:**
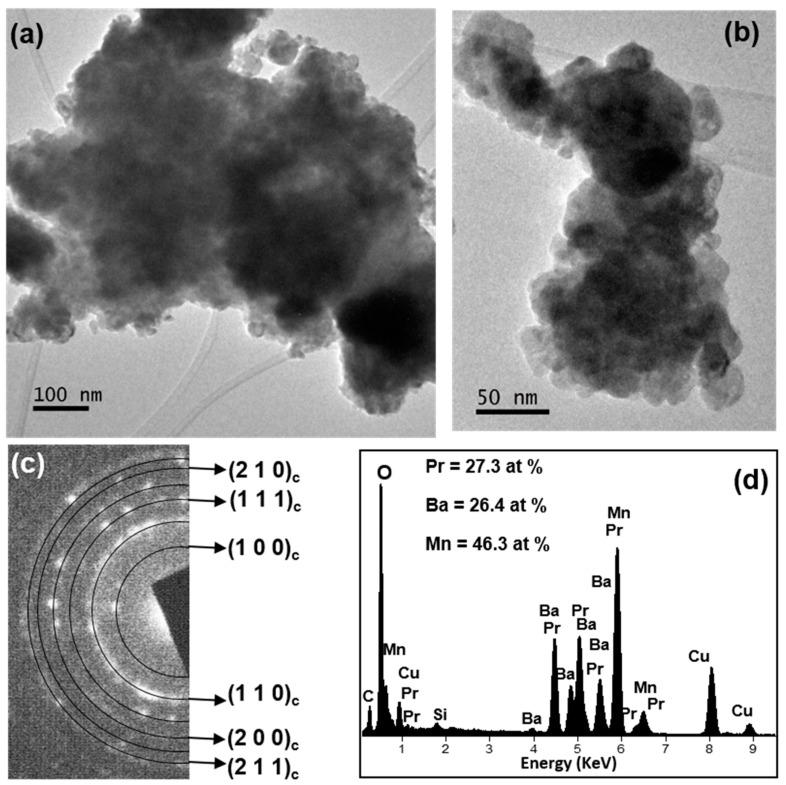
Microcharacterization of PBMO-m sample: (**a**,**b**) TEM images at different magnifications; (**c**) the corresponding selected area electron diffraction (SAED) pattern-rings and (**d**) summary results of the EDX measurements.

**Figure 8 nanomaterials-11-00380-f008:**
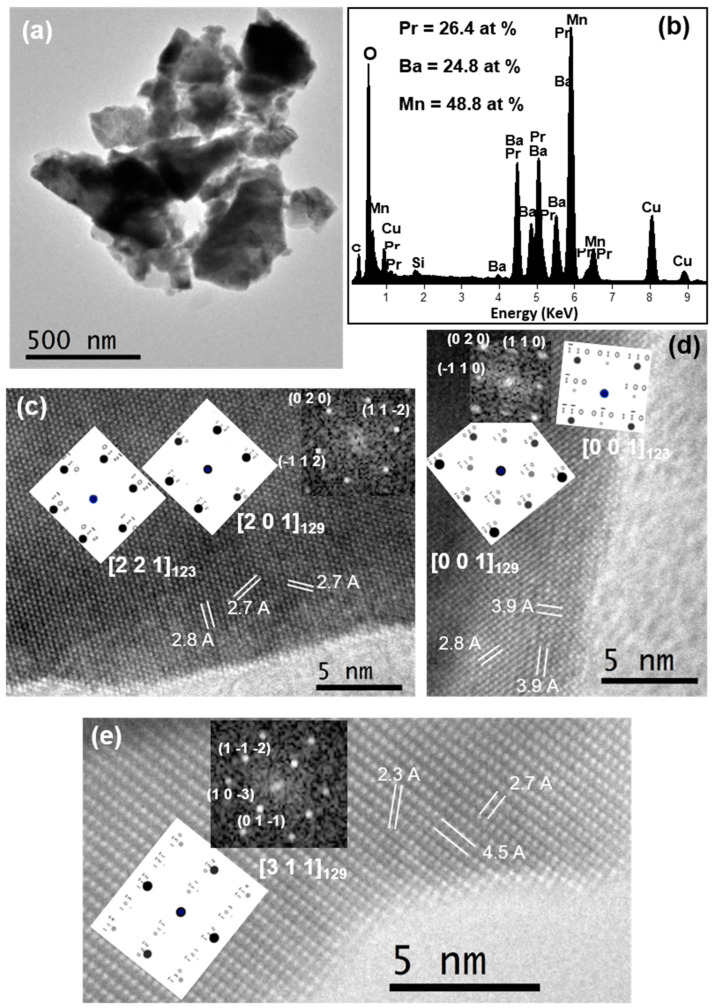
Microcharacterization of PBMO-m + 1150 °C/Ar sample: (**a**) TEM image; (**b**) representative EDX spectrum and the average atomic percentage of the sample; (**c**–**e**) HRTEM micrographs and the corresponding fast Fourier transform (FFT) and simulated ED patterns (insets).

**Figure 9 nanomaterials-11-00380-f009:**
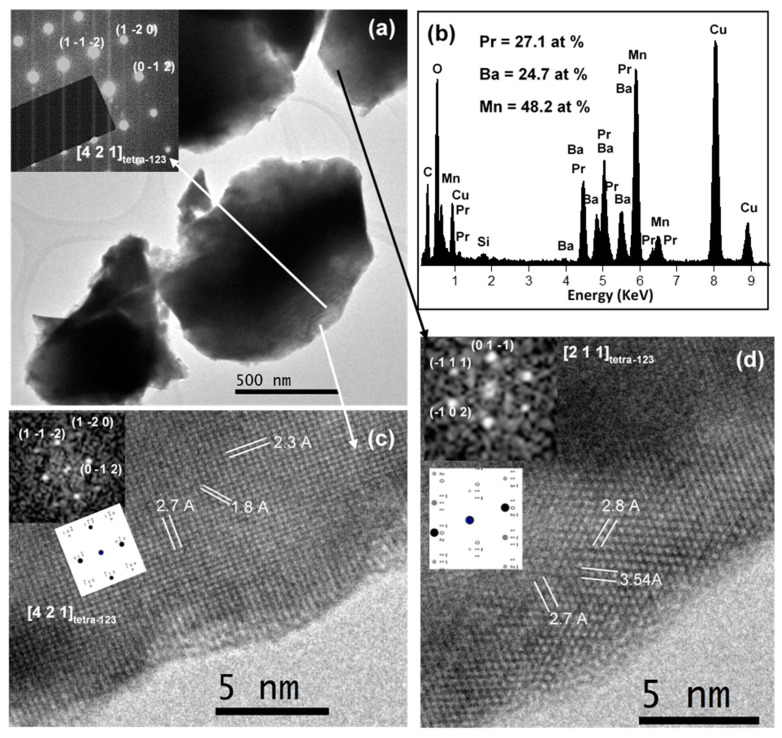
Microcharacterization of PBMO-m + 1150 °C/Ar + TG/air sample: (**a**) TEM image and a SAED pattern (inset); (**b**) representative EDX spectrum and the average atomic percentage of the sample; (**c**,**d**) HRTEM micrographs and the corresponding FFT and simulated ED patterns (insets).

**Figure 10 nanomaterials-11-00380-f010:**
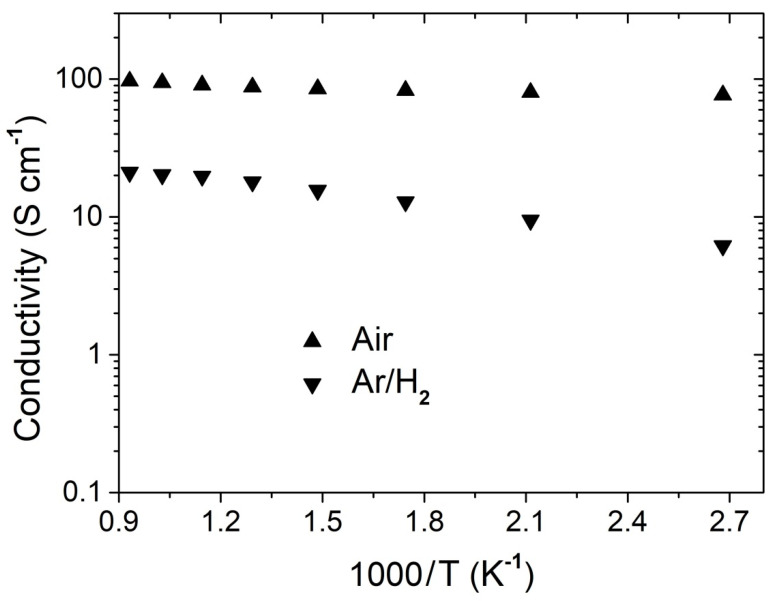
Temperature dependence of total conductivity of layered PBMO in air and argon.

## Data Availability

MDPI Research Data Policies at https://www.mdpi.com/ethics.

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
