# Peer review of "A Novel, Simple and Highly Efficient Route to Obtain PrBaMn2O5+δ Double Perovskite: Mechanochemical Synthesis"

_nanomaterials, 2021, doi:10.3390/nano11020380_

Round 1
Reviewer 1 Report
In this work by Garcia-Garcia and colleagues, the authors presented a convenient method for how PrBaMn2O5+δ double perovskite can be obtained by mechanochemical synthesis. The results are promising and fit the theme of the articles published by Nanomaterials. However, there are some issues, which must be clarified before the contribution can be accepted for publication. Please refer to the following comments:
1) Some details in the experimental part are missing. Examples:
- It would be good to describe the range of milling times in the first section as well rather than only in the results and discussion part
- gas flow rates for TGA measurements are not given
- the same is valid for impedance measurement
These are just examples, but there are more shortcomings like these. Please carefully go through this part and supplement all the necessary parameters to enable others to reproduce the study. Otherwise, it will be of limited impact as others will not be able to validate these findings and build on them.
2) The formatting of the article should be improved.
- many pages end with a lot of empty space, which should be populated with results
- plots are of different magnification which makes it difficult to compare the data
3) All XRD indices should be ascribed to the patterns in Fig. 1 (some are absent), Figs. 3a and 5 (all are not provided).
4) Please comment on the reproducibility of this process as it appears that the authors conducted the experiment just once. Did the authors try to repeat the processing and obtained the same results?
5) Plot generated from EDX software is hard to read. Please use the raw data and draw it yourself in a plotting software such as Origin. Besides, regarding Figs. 7d, 8b, and 9b, what are the uncertainty values for these results?
6) The lack of error analysis in Fig. 10 casts doubt if these results should be interpreted. Please conduct more measurements, gather enough data, produce the error bars, and add them to the plot.
7) Future outlook should be added to the Conclusions section.
Author Response
Reviewer 1
In this work by Garcia-Garcia and colleagues, the authors presented a convenient method for how PrBaMn2O5+δ double perovskite can be obtained by mechanochemical synthesis. The results are promising and fit the theme of the articles published by Nanomaterials. However, there are some issues, which must be clarified before the contribution can be accepted for publication. Please refer to the following comments:
1) Some details in the experimental part are missing. Examples:
- It would be good to describe the range of milling times in the first section as well rather than only in the results and discussion part
- gas flow rates for TGA measurements are not given
- the same is valid for impedance measurement
These are just examples, but there are more shortcomings like these. Please carefully go through this part and supplement all the necessary parameters to enable others to reproduce the study. Otherwise, it will be of limited impact as others will not be able to validate these findings and build on them.
More details on the experimental procedure used in the paper have been included in the Materials and Methods section.
2) The formatting of the article should be improved.
- many pages end with a lot of empty space, which should be populated with results
- plots are of different magnification which makes it difficult to compare the data
The manuscript has been rearranged in order to avoid empty spaces. Figures were also adjusted to a similar size.
3) All XRD indices should be ascribed to the patterns in Fig. 1 (some are absent), Figs. 3a and 5 (all are not provided).
In Figure 1, we provide the indices for the XRD lines of the final phase. We think it is not useful to provide those corresponding to the reactant phases and at intermediate milling times because this would make the figure less clear and hard to understand. We provide the indices in Figures 3 and 5 in the revised version.
4) Please comment on the reproducibility of this process as it appears that the authors conducted the experiment just once. Did the authors try to repeat the processing and obtained the same results?
In fact, we have performed the mechanochemical process several times and the final outcome was always the same. As you probably know, one of the characteristics of mechanochemistry is that it is highly reproducible. We have added this feature of mechanochemistry in the Introduction section.
5) Plot generated from EDX software is hard to read. Please use the raw data and draw it yourself in a plotting software such as Origin. Besides, regarding Figs. 7d, 8b, and 9b, what are the uncertainty values for these results?
The figures are changed in the revised version. In the text it is written that the maximum standard deviation is ±2.3%.
6) The lack of error analysis in Fig. 10 casts doubt if these results should be interpreted. Please conduct more measurements, gather enough data, produce the error bars, and add them to the plot.
The authors understand the reviewer point of view. However, we measured two pair of specimens (compacted discs), two for air and two for Ar/H2, which came from different preparations, but being prepared all under the same protocol, and we obtained almost the same curves for both of them, with a difference that can be considered negligible. Besides, and as far as we are concerned, it is not typical for such Arrhenius plots to be presented including error bars, as the reviewer can check on the vast majority of papers published. For these two reasons, we the error bars in Figure 10 are not included.
7) Future outlook should be added to the Conclusions section.
Future outlook has been added.

Reviewer 2 Report
The paper by Garcia-Garcia et al. describes the mechanochemical synthesis of PrBaMn2O5+δ double perovskite, reporting a wide range of characterizations, dedicated to the final compound, the oxygen non-stoichiometry and the intermediate steps of the synthesis. The synthesis leads to a pure phase, avoiding the formation of hexagonal byproduct, which can be easily reduced by high temperature annealing in Argon. The characterizations are convincing, the paper is well written and clear. I have just a few minor concerns:
line 42: I would remove “(ABO3)” as perovskites can have different formula, as the compound here investigated.
line 99: probably ‘of’ missing between “formation” and “which”
line 145: why semiquantification?
line 147: “using the tools offered by the X’Pert HighScore Plus software (version 3.0.5, PANalytical).” It is vague, which equations were used? Williamson Hall?
line 212: “a good fit of”. no fit is shown.
line 282 to 285: “Therefore, although the presence of BaCO3, a common and difficult to avoid impurity in BaO2, makes it difficult to analyze the thermogravimetric measurements of the PBMO-m sample, it is evident that the reduced layered structure (PrBaMn2O5+d) was obtained when PBMO-m was annealed in Ar; the use of a reducing atmosphere is not necessary.” I suggest rephrasing, the sentence is too long. I guess the authors mean that Ar is enough to reduce the perovskite, without recurring to H2 or reducing agents.
line 320: it is really hard to see whether the signal refers to a cubic or tetragonal phase. I would say pseudocubic.
I guess that black rings are a guide to the eye, but I can see more spots well out of the rings. What are they?
Author Response
Reviewer 2
The paper by Garcia-Garcia et al. describes the mechanochemical synthesis of PrBaMn2O5+δ double perovskite, reporting a wide range of characterizations, dedicated to the final compound, the oxygen non-stoichiometry and the intermediate steps of the synthesis. The synthesis leads to a pure phase, avoiding the formation of hexagonal byproduct, which can be easily reduced by high temperature annealing in Argon. The characterizations are convincing, the paper is well written and clear. I have just a few minor concerns:
line 42: I would remove “(ABO3)” as perovskites can have different formula, as the compound here investigated.
ABO3 is the general formula for perovskites and all related structures (doubles, layered, etc.) are derived from it.
line 99: probably ‘of’ missing between “formation” and “which”
The sentence has been corrected in the revised version
line 145: why semiquantification?
This has been modified in the revised version.
line 147: “using the tools offered by the X’Pert HighScore Plus software (version 3.0.5, PANalytical).” It is vague, which equations were used? Williamson Hall?
We have included more details in the revised version.
line 212: “a good fit of”. no fit is shown.
We have rewritten the sentence to make it clearer.
line 282 to 285: “Therefore, although the presence of BaCO3, a common and difficult to avoid impurity in BaO2, makes it difficult to analyze the thermogravimetric measurements of the PBMO-m sample, it is evident that the reduced layered structure (PrBaMn2O5+d) was obtained when PBMO-m was annealed in Ar; the use of a reducing atmosphere is not necessary.” I suggest rephrasing, the sentence is too long. I guess the authors mean that Ar is enough to reduce the perovskite, without recurring to H2 or reducing agents.
The sentence has been shortened in the revised version by removing unnecessary information about BaCO3.
line 320: it is really hard to see whether the signal refers to a cubic or tetragonal phase. I would say pseudocubic.
It is changed in the revised version.
I guess that black rings are a guide to the eye, but I can see more spots well out of the rings. What are they?
An ED-ring was not indexed with the corresponding (h k l) plane by mistake, now it is corrected and all the ED-rings are indexed.

Round 2
Reviewer 1 Report
Thank you. I am pleased with the conducted corrections. The paper can be published as is.